# *Akkermansia muciniphila* and *Faecalibacterium prausnitzii* in Immune-Related Diseases

**DOI:** 10.3390/microorganisms10122382

**Published:** 2022-11-30

**Authors:** Raden Mohamad Rendy Ariezal Effendi, Muhammad Anshory, Handono Kalim, Reiva Farah Dwiyana, Oki Suwarsa, Luba M. Pardo, Tamar E. C. Nijsten, Hok Bing Thio

**Affiliations:** 1Department of Dermatology, Erasmus MC, University Medical Center, 3015 GD Rotterdam, The Netherlands; 2Department of Dermatology and Venereology, Faculty of Medicine Universitas Padjadjaran—Dr. Hasan Sadikin Hospital, Bandung 45363, Indonesia; 3Department of Internal Medicine, Faculty of Medicine Universitas Brawijaya, Malang 65145, Indonesia

**Keywords:** *Akkermansia muciniphila*, atopic dermatitis, *Faecalibacterium prausnitzii*, human immunodeficiency virus, immune-related diseases, immunotherapy, probiotics, psoriasis, systemic lupus erythematosus

## Abstract

Probiotics and synbiotics are used to treat chronic illnesses due to their roles in immune system modulation and anti-inflammatory response. They have been shown to reduce inflammation in a number of immune-related disorders, including systemic lupus erythematosus (SLE), human immunodeficiency virus (HIV), and chronic inflammatory skin conditions such as psoriasis and atopic dermatitis (AD). *Akkermansia muciniphila* (*A. muciniphila*) and *Faecalibacterium prausnitzii* (*F. prausnitzii*) are two different types of bacteria that play a significant part in this function. It has been established that *Akkermansia* and *Faecalibacterium* are abundant in normal populations and have protective benefits on digestive health while also enhancing the immune system, metabolism, and gut barrier of the host. They have the potential to be a therapeutic target in diseases connected to the microbiota, such as immunological disorders and cancer immunotherapy. There has not been a review of the anti-inflammatory effects of *Akkermansia* and *Faecalibacterium*, particularly in immunological diseases. In this review, we highlight the most recent scientific findings regarding *A. muciniphila* and *F. prausnitzii* as two significant gut microbiota for microbiome alterations and seek to provide cutting-edge insight in terms of microbiome-targeted therapies as promising preventive and therapeutic tools in immune-related diseases and cancer immunotherapy.

## 1. Introduction

The term “probiotic” is derived from the Greek term “pro bios”, which means “for life”. It is associated with foods prepared from dairy, vegetables, and fruits that relied on fermentation [1]. Probiotics are living microorganisms that, when taken in adequate amounts, can have positive impacts on a person’s health [2,3]. Some probiotics are combined with prebiotics, which contain indigestible fiber that promotes the growth and activity of beneficial microorganisms. These combinations are called synbiotics, which are a combination of probiotics and prebiotics that play a role in balancing the microbiota in the digestive tract [2].

The concept of probiotics was first introduced in 1907; microbes that are consumed and digested have a good effect on individuals, especially for treating digestive tract diseases. The word probiotics was first used in 1965 to describe a substance secreted by one organism that can stimulate the growth of another organism [4]. Probiotics’ advantages are known to be caused by numerous mechanisms. These mechanisms are similar to how the gut microbiota affect health. Firstly, probiotics stop harmful bacteria from invading the digestive system. Secondly, probiotics are known to boost the function of the digestive tract’s mucosal barrier. Thirdly, probiotics can modify the immune system to stop it from overreacting to inflammation. Finally, probiotics produce and release metabolites with beneficial and anti-inflammatory properties. The enteric nervous system and the central nervous system can both be modulated by probiotics [3].

There may be one or more bacterial strains present in probiotic products. In contrast to probiotics, prebiotics are components of food that cannot be digested and are mostly in the form of fiber. Prebiotics have the advantage of specifically stimulating the activity and proliferation of intestinal microorganisms, which has a positive impact on health [5,6]. Synbiotic refers to a product that contains both prebiotics and probiotics [6]. The antioxidants required to reduce oxidative stress might be found in probiotics. These promote the body’s efforts to lessen the major cause of several chronic human diseases. Probiotics and synbiotics are used to treat chronic illnesses due to their role in immune system modulation and anti-inflammatory response [7].

Due to their impact on the gut microbiota, a variety of microbial species have been receiving increased attention. Since many disorders have been found to be closely related to gut microbiota, it is crucial to enhance the host’s health by altering the intestinal makeup [8]. Dysbiosis, or abnormal changes in the microbiome’s composition, can be managed through probiotics. The intake of advantageous bacterial strains can lessen harmful bacteria, improve microbial dysbiosis, and alter immune function by delivering exogenous bacteria and promoting either temporary or permanent colonization [9]. Intestinal dysbiosis is linked to metabolic disorders, gastrointestinal infections, and inflammatory bowel diseases [10].

*Faecalibacterium prausnitzii* (*F. prausnitzii*) is an important member of the Firmicutes phylum and is one of the frequent bacteria in the normal human microbiota. It is currently known that *F. prausnitzii* can account for up to 5% of the entire fecal microbiota in healthy individuals. Furthermore, it has been proposed that this bacterium acts as both a sensor and an activator in human intestinal health [11]. In order to preserve health, this species, which produces butyrate, influences homeostasis and physiological processes [12]. Its anti-inflammatory properties are widely known, and changes in the composition of *F. prausnitzii* in the gut are thought to be related to a number of human diseases [13].

*Akkermansia muciniphila* (*A. muciniphila*), an abundant symbiont in the human gut, recently raised concerns among scientists and clinicians around the world. According to normal circumstances, the population of *A. muciniphila* makes up between 3% and 5% of the intestinal species in an adult colon and more than 1% of the entire microbiota in the feces, indicating that it is one of the most prevalent bacteria in the microbial community [14]. The Gram-negative bacterium *A. muciniphila*, which colonizes in the mucosal layer, is thought to be a viable probiotic candidate. It is well-recognized that *A. muciniphila* has a significant role in enhancing the host’s immune system and metabolic processes [15]. It can control how the host’s metabolism, immune system, and intestinal barrier work by producing short-chain fatty acids (SCFAs), primarily propionate, and acetate [16]. *A. muciniphila* was discovered to be adversely connected with numerous disease states and positively correlated with a healthy intestine. Additionally, *A. muciniphila* is said to control immunological function, which enhances the synthesis of antimicrobial peptides and improves gut homeostasis [17].

Some studies have identified intestinal microbiota in inflammatory bowel disease (IBD) compared to healthy controls. A common outcome of IBD is a decrease in microbial diversity [18]. The results are more erratic when it comes to microbiota composition. However, it appears that a lesser abundance of bacteria that produce butyrate or propionate, such as *F. prausnitzii* [19] and *A. muciniphila* [20,21], is a common characteristic. This finding emphasizes the importance of these microbes as probiotics [22].

The utilization of commensal microorganisms as possible probiotic agents is becoming more and more popular. The most obvious reason, among many others, is that in the last ten years, the involvement of commensal bacteria in homeostatic crosstalk has begun to be revealed [11]. In addition, probiotics have been shown to have positive benefits in studies on animal models of cancer formation and mucosal inflammation, but there is a dearth of clinical evidence to support these claims in humans [23]. These innovative experiences have enabled researchers to posit new directions for probiotic use; in particular, probiotics have been suggested as a way to improve the response to immunotherapy.

A review of the anti-inflammatory effect of *A. muciniphila* and *F. prausnitzii*, especially in immune-related diseases, has not yet been conducted. In this paper, we would like to elaborate on this information, which will be useful for further research in the field. We specifically review the anti-inflammatory effects of probiotics in several immune-related diseases, namely human immunodeficiency virus (HIV), systemic lupus erythematosus (SLE), and chronic skin diseases such as psoriasis and atopic dermatitis (AD), and in response to immunotherapy.

## 2. Overview

### 2.1. Akkermansia Muciniphila

*Akkermansia* is a Gram-negative, oval-shaped, non-motile, and oxygen-tolerant anaerobic bacterium (Table 1). It is a member of the Verrucomicrobia (phylum), Verrucomicrobiae (class), Verrucomicrobiales (order), Verrucomicrobiaceae (family), Akkermansia (genus), and muciniphila (species) [14]. It is found in mice, hamsters, wild animals, and humans [24]. In the guts of both humans and animals, Derrien et al. discovered the Verrucomicrobia phylum member *A. muciniphila* (type strain Muc^T^; ATCC BAA-835). Despite having only been discovered recently, *A. muciniphila* has been the subject of in-depth research about how it affects human metabolism. Its impact on chronic conditions, such as immune diseases and cancer, has been investigated.

The majority of *A. muciniphila* distribution is in the distal regions of the small and large intestines, where it uses mucin as its primary energy source to produce the amino acids and sugar groups necessary for bacterial development. In sterile mice, *A. muciniphila* was effectively colonized in the caecum, where the level of colonization was the highest in vivo. This might be explained by the fact that the caecum produced the majority of the mucin [25]. In addition to producing organic acids, such as acetate and propionate, when *A. muciniphila* breaks down mucin, it also releases less complicated carbohydrates from the mucin layer [26]. *A. muciniphila* has created new opportunities for the use of next-generation therapeutic probiotics because of its distinctive function, high universality, and richness in nearly all life stages [26,27].

It has been discovered to be prevalent in both healthy adults’ and children’s intestines, making up between 1 and 4% of the total gut microbiota from an early age. The abundance level can rise by up to 5% in unusual circumstances [28]. Its level of abundance is essential for normal physiological processes, and any abnormalities in these levels are closely related to the pathophysiology of chronic diseases [29]. *A. muciniphila* may spread from mothers to newborns through human milk, which would account for its presence in the digestive tract of newborn infants [30]. This bacterium can establish a stable colony in the human gut within a year of birth, and it eventually achieves the same level of abundance as in the guts of healthy people, although it gradually declines in the elderly [28,30]. According to reports, *A. muciniphila* may maintain the balance of the host’s gut microbes by transforming mucin into advantageous byproducts [28]. There is currently no information that *A. muciniphila* alone is pathogenic, but it is unknown if it can cause disease when combined with other bacteria.

Dendritic cells that pass through the intestinal epithelium acquire commensal bacteria from the gut lumen and reach the bloodstream, where they can survive for days. Their translocation to the mammary glands has recently been documented. *A. muciniphila* tend to be linked to proinflammatory signals, such as greater TNF-α and IFN-γ in colostrum and lower IL-10 and IL-4 levels during lactation [31]. *A. muciniphila* colonization in sterile mice did not result in negative side effects or the upregulation of pro-inflammatory cytokine levels [25]. *A. muciniphila* have been positively correlated with a healthy gut [20,32]. Due to their ability to effectively utilize the mucin in the gastrointestinal tract, *A. muciniphila* are regarded as potential probiotics [15]. They have a distinctive method of surviving, namely the release of carbon and nitrogen resources as a result of the host’s gastrointestinal mucin degrading [33]. As a result, we suggest that *A. muciniphila* supplementation can be considered safe and reasonable.

Recently, it has been thought that *A. muciniphila* play a key role in both pathogenic and homeostatic aspects of human physiology. The connections between the prevalence of *A. muciniphila* and various disorders and diseases have been the subject of several human and animal investigations.

**Table 1 microorganisms-10-02382-t001:** Characteristics of *A. muciniphila* and *F. prausnitzii*.

	*A. muciniphila*	*F. prausnitzii*
**Microbiologic**	Gram-negative oxygen-tolerant anaerobic	Gram-positive strict anaerobe
**Shape**	Oval shape	Rod shape
**Phylum/Class**	Phylum: Verrucomicrobiota Class: Verrucomicrobiae	Phylum: Bacillota Class: Clostridia
**Discovered**	2004 [26]	2002 [34]
**Typical features**	Produces organic acids such as acetate and propionate when it breaks down mucin Releases less complicated carbohydrates from the mucin layer Degrades human milk oligosaccharides in newborn infants’ stomachs	Produces butyrate and other short-chain fatty acids through the fermentation of dietary fiber Elicits a tolerogenic cytokine profile and has been linked to additional anti-inflammatory capabilities Their supernatant reduces the intensity of inflammation by releasing metabolites that improve intestinal barrier performance and have an impact on paracellular permeability
**Immunologic** **features**	Decrease in the anti-inflammatory cytokines IL-10 and IL-4 Rise in the pro-inflammatory cytokines TNF-α and IFN-γ Colonization did not result in negative side effects or an upregulation of pro-inflammatory cytokine levels	Increase in very low secretion of pro-inflammatory cytokines such as IL-12 and IFN-γ and enhanced secretion of the anti-inflammatory cytokine IL-10 Suppresses the NF-κB pathway utilizing the NF-κB-luciferase

### 2.2. Faecalibacerium Prausnitzii

Gram-positive, mesophilic, rod-shaped, commensal strict anaerobe *F. prausnitzii*, also known as “fecomucus” bacteria (Table 1), was identified from the human microbiome. As one of the most prevalent butyrate-producing species, it makes up about 5% of the fecal microbiota [35]. *F. prausnitzii* has been frequently demonstrated to be one of the main butyrate producers in the intestine. Butyrate is essential for the health of the host and the physiology of the gut. It serves as the colonocytes’ primary source of energy and has anti-colorectal cancer (CRC) and anti-IBD effects. This species’ potential to elicit a tolerogenic cytokine profile has been linked to additional anti-inflammatory capabilities. *F. prausnitzii* produces metabolites; one of them is butyrate, which induces a very low secretion of pro-inflammatory cytokines, such as IL-12 and IFN-γ, and an enhanced secretion of the anti-inflammatory cytokine IL-10 in human PBMC culture supernatant and colitis model mouse blood serum [36,37]. Additionally, it has been demonstrated that *F. prausnitzii* supernatant reduces the intensity of inflammation by releasing metabolites that improve intestinal barrier performance and have an impact on paracellular permeability [13].

Since it was discovered that Faecalibacterium produces butyrate, it is also a possible probiotic candidate. This brand new anti-inflammatory bacterium could be particularly helpful as a preventative measure against inflammatory bowel illness [38]. In a previous study, it was discovered that administering *Enterococcus durans* EP1 increased the levels of *F. prausnitzii* [39].

*F. prausnitzii* has been subjected to a number of animal tests to assess its potential as a probiotic. According to Qiu et al., *F. prausnitzii*, together with its supernatant (which contains some unidentified metabolites), delivered orally to model mice reduced the severity of chemically induced colitis by controlling the differentiation of the inflammatory and tolerogenic T cell subsets and the excretion of the necessary cytokines. They also demonstrated that *F. prausnitzii* encouraged the expansion of Treg in splenocytes and peripheral blood cells, which was possibly helped by butyrate properties [36,37,40]. Another study supported this idea by demonstrating how the *F. prausnitzii*-derived microbial anti-inflammatory molecule (MAM), a bioactive peptide produced by *F. prausnitzii* and derived from a single 15 kDa protein (ZP05614546.1), suppressed the NF-κB pathway utilizing the NF-κB-luciferase in vivo in a mice model of induced colitis [41]. Their findings made it evident that the DNBS-activated NF-κB pathway was diminished when MAM cDNA was delivered to the intestinal mucosa [42].

## 3. Immune Hyperactive Disease

### 3.1. Systemic Lupus Erythematosus

Some studies on the microbiome in lupus patients have discovered microbial dysbiosis in this group of patients. Various cohorts have shown lower Firmicutes-to-Bacteroidetes ratios and gut microbiota richness and diversity [43,44]. Segmented filamentous bacteria have been found to be more prevalent in lupus patients’ gut microbiomes. While the prevalence of Bifidobacterium is inversely correlated with lupus activity, that of *Streptococcus*, *Campylobacter*, *Veillonella*, *Clostridiacae*, and *Lachnospiraceae* is favorably correlated with SLE disease activity [45,46]. Finally, the guts of lupus patients are richer in pathological microbiomes, such as *Ruminococcus gnavus* and *Enterococcus gallinarum* [47,48]. Another study found that some anti-inflammatory bacteria, particularly those from the genera *Roseburia, Faecalibacterium,* and *Bifidobacterium,* decreased, while some pro-inflammatory bacteria, particularly those from the genera *Streptococcus* and *Campylobacter,* expanded in the feces of SLE patients, particularly the active ones. This led to the release of inflammatory factors, which in turn increased the level of systemic inflammation [45].

The influence of SLE medication on the gut microbiome has also been explored in some studies. Patients with SLE may experience a leaky gut as a direct side effect of their drugs or as a result of infections brought on by immunosuppression. For instance, NSAIDs can cause intestinal epithelial cells to experience oxidative stress, mitochondrial damage, and endoplasmic reticulum damage, which increases gut permeability and localized inflammation [49]. Glucocorticoids, a class of medications commonly used to treat lupus, and the hydroxychloroquine treatment of SLE patients both reduced the diversity of microorganisms [50]. Tacrolimus, a calcineurin inhibitor, by decreasing mitochondrial function in gut epithelial cells, causes gut leakage in both humans and rats [51]. Finally, immunosuppressive medication therapy raises SLE patients’ chances for bacterial and viral infections. Older patients, men, people of color (Black and Indigenous), those who used glucocorticoids or immunosuppressants, and older patients were among those at higher risk for a first serious hospitalized infection. However, those who used hydroxychloroquine were less likely to become ill. Similar to the healthy population, the majority of infections are caused by bacteria. The bloodstream, skin, soft tissue, and gastrointestinal, urinary, and respiratory tracts are often infected areas. *S. pneumoniae*, *E. coli*, and *S. aureus* are some of the pathogens that are most frequently isolated. Herpes zoster, cytomegalovirus (CMV), and human papillomavirus (HPV) are typical viral infections [52,53]. Intestinal infections brought on by these medications may cause immune activation, inflammation, and disruption to the gut barrier in the hosts.

Several strategies to tackle this include diet and lifestyle modification, the use of probiotics, and some medical treatments [47,54]. The use of probiotics or prebiotics was shown to give benefits in lupus patients. Probiotics can control the level of inflammation and lower the generation of autoantibodies, which can diminish the severity of lupus [55]. Additionally, numerous investigations on the benefits of probiotics in SLE patients and mice models have been conducted. In summary, there are several benefits of probiotics in SLE, and they can help alleviate disease symptoms. Probiotics may be helpful in reducing inflammation and suppressing inflammatory reactions, according to studies on both animals and humans. Certain probiotic bacteria, such as *Bifidobacterium bifidum*, *Lactobacillus*, *Ruminococcus obeum*, and *Blautia coccoides*, have been shown in studies using the SLE animal model to help control excessive inflammation and reestablish tolerances [56].

Another study showed that patients’ serum levels of C3 were inversely linked with the prevalence of *Ruminococcus*, *Bacteroides*, and *Akkermansia* in stool. It is clear that *Akkermansia* spp. contribute to improved gut barrier function and host immunological homeostasis in the gut mucosa [57]. This discrepancy shows that even though in most studies, *A. muciniphila* demonstrated an anti-inflammatory effect, in this particular study, it proved otherwise, and thus, further analysis is needed.

There have been several studies discussing the role of *Faecalibacteria*. According to one study, *Faecalibacterium* has a positive link with several metabolites, including pentanoate, hippurinate, succinate, and lactic acid, while it has a negative correlation with glycolic acid. Compared to healthy controls, SLE patients’ gut microbiomes were less diverse. The taxonomic class *Cryptophyta* and the genus *Roseburia* were reduced in the gut microbiota of the SLE group, while the genus *Faecalibacterium* and its species *prausnitzii* showed significant decreases in patients with SLE. This resulted in considerably lower levels of pentanoate, an SCFA with the ability to suppress autoimmunity. Additionally, there was a drop in the concentrations of D-lactic acid, succinate, and hippurate, which were said to be byproducts of microbial carbohydrate fermentation. Furthermore, there was a substantial increase in glycolic acid, which has been shown to cause renal damage. These findings imply the potential effects of *Faecalibacterium* as a candidate for probiotics in SLE patients [46].

### 3.2. Atopic Dermatitis

Along with the skin microbiota, the gut microbiota are also strongly linked to atopic diseases such as AD. According to recent research, the development of AD may be associated with altered gut microbiota. It is known that the gut microbiota affect host immune responses at a systemic level. Several investigations have shown that AD is linked to intestinal dysbiosis, particularly in the early years of life. Infantile gut dysbiosis may be able to predict the onset of AD, but childhood AD history leaves long-lasting evidence in the gut microbiota. Neonates with childhood AD experience have shown lower levels of *Bifidobacteria*, *Akkermansia*, and *Faecalibacteria* than their healthy counterparts [58].

Children with atopic diseases had lower levels of *A. muciniphila*, which suggests that it is crucial for the development of IgE-related atopic diseases. *A. muciniphila* may interact with intestinal epithelial cells to release IL-8 for immunomodulatory effects, according to the link between the immunological response and the low levels of the organism in atopic children [59].

According to recent research, *A. muciniphila* and *F. prausnitzii* were less prevalent in the gut microbiota of children with asthma and AD. These bacteria’s released metabolites may cause anti-inflammatory cytokines and stop pro-inflammatory cytokines from being produced [60]. However, the clinical implications of these bacteria are still not well understood.

Based on a study conducted by Song et al., a metagenomic analysis of fecal samples from AD patients revealed a substantial drop in *F. prausnitzii* species as compared to control patients. The AD patients also had lower levels of fecal SCFAs, particularly butyrate. This was probably due to a reduction in high butyrate and propionate producers, particularly those connected to *F. prausnitzii* strain A2-165, whose absence has been linked to Crohn’s disease patients, as a result of an intraspecies compositional change in *F. prausnitzii* [61].

Specific *A. muciniphila* and *F. prausnitzii* strains can be administered to patients to greatly reduce AD symptoms by modulating the immune system and gut barrier function. It is believed that *A. muciniphila* and *F. prausnitzii*’s anti-inflammatory actions are brought on by their metabolites, which influence the genes that control gut function, particularly in host intestinal epithelial cells, by secreting SCFAs such as propionate, butyrate, and acetate. In contrast to *A. muciniphila*, which produced acetate and propionate, *F. prausnitzii* produced butyrate as a significant metabolite [62]. *F. prausnitzii* cannot be grown commercially due to its high oxygen sensitivity, and therefore, Koga et al. gave kestose to children with AD in an effort to increase the *F. prausnitzii* population and discovered improvements in AD symptoms in 2–5-year-old infants, which was associated with an elevation in their fecal *F. prausnitzii* composition [63].

In an animal model investigation, *A. muciniphila* EB-AMDK19 or *F. prausnitzii* EB-FPDK11, isolated from humans, were given orally to 2,5-dinitrochlorobenzene (DNCB)-induced AD models utilizing NC/Nga mice at a daily dose of 108 CFUs/mouse for six weeks in order to assess their potential therapeutic effects on AD. Each strain of *A. muciniphila* and *F. prausnitzii* was administered as a consequence, and this improved AD-related parameters such as the dermatitis score, scratching behavior, and serum immunoglobulin E levels. Additionally, the *A. muciniphila* and *F. prausnitzii* therapies addressed the imbalance between the T helper (Th)1 and Th2 immune responses brought on by DNCB by reducing the level of thymic stromal lymphopoietin (TSLP), which causes the production of Th2 cytokines. As functions were being restored, the oral administration of the bacteria increased the production of filaggrin in the epidermis and ZO1 in the gut barrier by secreting SCFAs such as propionate, acetate, and butyrate. When compared to dexamethasone use for AD, EB-FPDK11 and EB-AMDK19 appeared to have comparable or higher efficacies in terms of their anti-atopic actions. As a result, supplementation with *A. muciniphila* EB-AMDK19 or *F. prausnitzii* EB-FPDK11 may provide a cutting-edge therapeutic option for AD patients. However, in order to establish the effectiveness of EB-FPDK11 and EB-AMDK19 in humans, clinical studies are required [62].

Selective microbiota modulation as an AD treatment has received much attention because dysbiosis is correlated with AD. By utilizing probiotics, prebiotics, or synbiotics, the gut microbiota can be normalized [2]. The dysbiosis condition can be reversed with the use of probiotics.

### 3.3. Psoriasis

The diversity of the gut microbiome can have a substantial impact on the development of the immune system and disease risk, particularly for autoimmune diseases such as psoriasis. A study using microbiota and inflammation-related characteristics suggested that microbiota dysbiosis may result in an abnormal immune response in psoriasis. By generating butyrates with antioxidant properties, *F. prausnitzii* contribute significantly to gut homeostasis. NF-κB, which controls the inflammatory response, is inhibited in intestinal epithelial cells by 15 kDa protein as an anti-inflammatory protein produced by *F. prausnitzii*, and they also provide intestinal membrane-lining cells (enterocytes) [40,64]. These studies have emphasized the critical notion that the microbiome is involved in maintaining a healthy gut barrier.

In comparison to the control group, the psoriasis group showed a decline in the *Lachnospira* and *A. muciniphila* species, according to a Brazilian study that examined the composition and diversity of the gut microbiota in 21 people with psoriasis [65]. It was discovered that psoriasis patients showed a significant decrease in *A. muciniphila* in a study by Tan et al. that examined the microbial composition of healthy controls and patients with psoriasis vulgaris. The studies conducted by both Scher et al. and Eppinga et al. found a decline in the *F. prausnitzii* composition. The drop in the composition of *A. muciniphila* and *F. prausnitzii* had an impact on psoriasis [66,67] because they are considered to be helpful microbes that produce SCFAs, which are protective against systemic inflammatory diseases and are essential for preserving the integrity of the gut epithelium [68].

*Pseudobutyrivibrio*, *Ruminococcus*, and *Akkermansia* were reported to be less common in both psoriatic and psoriatic arthritis patients by Scher et al. [69]. It was demonstrated how the diversity gradually declines along with the onset of joint illness in psoriasis patients [70]. Despite the fact that the gut microbiota of psoriatic arthritis differs from psoriasis, these alterations in the gut microbiome are actually comparable with IBD, one of psoriasis’s comorbidities [71]. In the study conducted by Tan et al., it was found that the abundance of *A. muciniphila* was markedly decreased in psoriasis patients, a finding that was also similar for patients with IBD and obesity, offering a unique perspective on the pathophysiology of psoriasis. A study indicated that because *A. muciniphila* is inversely correlated with cardiometabolic illnesses, diabetes, obesity, and low-grade inflammation, it may have an impact on the course and severity of psoriasis [27].

There is a balance between the development of naive CD4+T cells into effector T cells [Th1, Th2, and Th17] and regulatory T cells [Tregs] in the healthy gut microbiota. It is thought that the Th1 and Th17 arms of the adaptive immune system are predominate in autoimmune disorders like Crohn’s disease and psoriasis [66]. Psoriasis frequently develops in conjunction with gastrointestinal inflammation, such as IBD. IL-17A is the main factor causing skin pathology in psoriasis patients, and serum beta defensin-2 is a simple-to-measure biomarker of IL-17A-driven skin pathology. Additionally, the expression of IL-17A in the mucosa was increased in IBD patients. It is believed that altered immunological and inflammatory responses in the intestinal mucosa are related to IL-17 expression in psoriasis and also in IBD. All things considered, these studies revealed that the proportion of intestinal microbiota, including *A. muciniphila*, is altered in psoriasis, which offers new information about how the human intestinal microbiota contributes to the etiology of the disease.

The administration of probiotic *Lactobacillus* strains has been shown to improve psoriasis by modulating the gut microbiota. Probiotics have been demonstrated to be beneficial for treating psoriasis. However, there is still room for development in this area due to differences in methods and probiotic formulations [72]. To further understand the function of *A. muciniphila* in the pathogenesis of psoriasis, future research must be conducted to look at the possibility of preventing or treating psoriasis through the gut microbiota [67]. Treatment for psoriasis may benefit from modifying the gut microenvironment with probiotics if the psoriasis disease is mediated by the gut microbiota. Numerous researchers have shown interest in treatment strategies with probiotic, prebiotic, and synbiotic supplements among psoriasis patients; as a result, numerous investigations are currently addressing this topic in both experimental and clinical research.

## 4. Immunodeficiency

### Human Immunodeficiency Virus

A healthy gut microbiota depends on the innate and adaptive immune systems [73]. Though the exact mechanisms of immune interactions with gut microbes are still being explored, research has demonstrated that the transplantation of T-regs to T cells in animal models may restore gut microbial diversity and that immunoglobulin A (IgA) binding to commensal bacteria may change microbial gene expression [74]. Therefore, the B cell generation of antibodies and CD4+ T cell immunological control may be crucial for preserving a favorable gut microbial ecology [75]. As a result, it should not come as a surprise that the quick and severe loss of intestinal immunity brought on by human immunodeficiency virus (HIV) infection, particularly the loss of CD4+ T cells, would have a significant impact on the gut microbiome. The schematic picture of how HIV and gut microbial dysbiosis affect each other can be seen in Figure 1.

Although several studies have shown the benefit of *Akkermansia* as anti-inflammatory probiotics, there have also been several studies that have proven otherwise. A study in children living with HIV showed an increase in the abundance of *A. muciniphila* compared to the normal population [76]. These outcomes may influence the justification of co-trimoxazole usage as a prophylactic in this population. The observed loss of diversity is either caused by antiretroviral therapy (ART) and/or HIV infection, as evidenced by the current body of literature, or is caused by the use of co-trimoxazole during very early life, a critical period for microbiome development.

*Akkermansia* relative abundance is also shown to decrease in mice treated with tuftsin and phosphorylcholine (TPC, an immunomodulatory substance derived from helminth), which indicates a better disease process [77]. It was discovered that *A. muciniphila* activates TLR4 and TLR2 via the NF-*κ*B pathway. The lipopolysaccharide (LPS) from *A. muciniphila*, which is probably the trigger for TLR2 and TLR4, caused peripheral blood mononuclear cells (PBMC) to produce IL-8 and IL-6 and trace levels of IL-10 and TNF-α. Purified recombinant Amuc 1100, an outer membrane pilus-like protein, may specifically activate TLR2 and also cause PBMCs to produce IL-1, IL-6, IL-8, IL-10, and TNF-α. Extracellular vesicles (EV) produced from *A. muciniphila* emitted increasing levels of IL-6 in a dose-dependent manner in another investigation [78]. While some studies have indicated that *A. muciniphila* reduces colon inflammation, others have demonstrated that this bacterium actually worsens the condition of gut inflammation by degrading the mucin, which makes it easier for bacteria and luminal antigens to enter the interior layers of the colon [79,80,81,82,83] and is even related with certain cancer types [84]. It is hoped that a deeper understanding of the intricate host–microbiota interactions will clarify this disparity, which is still unclear at this time.

Inflammation, microbial translocation, and damage to the epithelium of the gut are all thought to be common determining mediators of inflammatory non-AIDS comorbidities in people living with HIV. The “sentinel of the gut” is *A. muciniphila*, which has been demonstrated to support gut barrier integrity, alter the immunological response, reduce inflammation, and enrich butyrate-producing bacteria. It has been demonstrated that *A. muciniphila* supplementation and other methods that encourage the abundance of *A. muciniphila* are effective treatments for several cancers and metabolic disorders. We propose that a gut microbiota enriched in *A. muciniphila* can reduce microbial translocation and inflammation, lowering the risk of developing non-AIDS comorbidities and enhancing the quality of life in PLWH. Recently, clinical trials involving metformin, prebiotics (CIHR/CTN NCT04058392), or fecal microbiota transplant (FMT) to increase *A. muciniphila* abundance have come to fruition [85]. Together, these findings may help to partially explain the association between gut health and *A. muciniphila* levels [17].

Probiotic supplementation as a companion to combination antiretroviral therapy (cART) may expedite the restoration of gut immunity and barrier function while lowering the systemic immunological activation brought on by bacterial translocation, according to persuasive evidence from in vitro and non-human primate models. PROOV IT I and II studies, studying the ability of the microbiome to enhance surrogate measures of HIV morbidity and mortality among people using cART, showed that although the participants reported that the probiotic and regular ART treatment plan was safe and well tolerated, it had no effect on bloodstream inflammation or adverse immune activation. It also had no discernible effects on the operation of the gut immune system. Probiotic supplementation may have additional health advantages, but according to that study, there is no need further study whether this probiotic can reduce detrimental inflammation in immunological non-responders who are currently receiving effective ART [86].

Numerous studies have noted the *Faecalibacterium* pattern in HIV patients. In HIV-negative patients, the Firmicutes species *Faecalibacterium* and *Subdoligranulum* predominated; in HIV-positive patients, *Faecalibacterium* was depleted and the Proteobacteria species *Stenotrophomonas* and *Achromobacter*, as well as *Subdoligranulum*, appeared plentiful. This finding suggests that important commensals have disappeared in HIV-infected individuals [87]. According to certain data, there was a significant difference at the genus level between the fecal microbiome compositions of the chronic HIV-infected patients and the non-HIV-infected controls. The changes in relative abundance were most noticeable for the genera *Phascolarctobacterium*, *Faecalibacterium*, *Roseburia*, and *Lachnospira*, as well as for taxa within the *Veillonellaceae*, *Ruminococcaceae*, and *Lachnospiraceae* families in the order *Clostridiales*, class Clostridia within Firmicutes, a phylum predominated by Gram-positive bacteria. This study emphasized how the distinct microbiome shift in Chinese HIV-infected patients affected the course of the illness and how well they responded to ART. Improved and comprehensive care of this condition will result from better knowledge about the relationships between the pro-inflammatory bacterial community composition of the human fecal bacterial microbiome and chronic HIV infection–related health problems [88].

Another study described how highly active antiretroviral therapy (HAART) affected the microbiome in HIV patients. It was discovered that patients receiving HAART had significantly higher concentrations of the genera *Prevotella*, *Faecalibacterium*, *Alistipes*, *Oscillibacter*, *Barnesiella*, *Dialister*, and *Odoribacter*, whereas patients not receiving HAART had higher concentrations of the genera *Megamonas*, *Veillonella*, *Blautia*, *Clostridium* XVIII, and *Enterococcus*. Although the viral load is reportedly reduced by the HAART treatment for HIV patients, the Firmicutes/Bacteroidetes ratio remained much greater than that of the untreated healthy controls. As a result, although being an effective treatment, HAART did not entirely restore the fecal microbiome of the HIV-infected patients [89].

## 5. Cancer Immunotherapy

It is becoming more widely recognized that the gut microbiota affects host immunity and the therapeutic effects of cancer treatment. This is significant since altering microbiota can be achieved through a variety of methods, opening up new perspectives for the treatment of cancer. Immunotherapy for cancer is greatly affected by gut microbiota [90]. In order to increase the effectiveness of cancer immunotherapy and lower associated side effects, the modulation of gut microbiota was recommended as a new approach [91]. Currently, a number of interventional studies are looking into the viability of altering the microbiota to enhance therapeutic efficacy and lessen drug-induced toxicity. In particular, the microbiota can be altered by a variety of methods, most notably through the successful use of pre- or probiotics or by fecal microbiota transplantation in recent years [92]. Several areas of research focus on how microbiota can improve the effectiveness of immunotherapy. In fact, multiple studies have indicated that the composition of the gut microbiota affects the effectiveness of anti-PD-1 therapy. More specifically, a number of microbiota signatures were suggested as prospective biomarkers of immunotherapy efficacy in melanoma [93].

In order to treat melanoma, researchers specifically investigated how prebiotics can increase bacterial taxa that enhance anti-tumor immunity in a mouse model. More specifically, the data showed that insulin played a role in the regulation of tumor growth, improving the effectiveness of immunotherapy against melanoma while delaying the emergence of drug resistance. In addition, the data showed that the gut microbiota was necessary for the development of an immune response-supporting anti-cancer treatment. The results of this investigation should be confirmed by patient-centered clinical studies, but they serve as a foundation for future clinical trials [94].

According to Matson et al., four patients with metastatic melanoma who had clinical responses to anti-PD-1–based immunotherapy had high levels of *A. muciniphila* [95]. A mouse melanoma model showed increased tumor control and improved immunotherapy efficacy after being gavaged with fecal material from responding patient donors. Additionally, patients with melanoma who responded to the PD-1 blockade medication had increased levels of beneficial gut microbiota, according to Gopalakrishnan et al. [96]. These results showed that cancer immunotherapy paired with *A. muciniphila*, as one of the crucial probiotics in selective microbiota transplantation, is predicted to improve clinical outcomes for patients in the near future [97].

The effects of tumor immunotherapy are significantly correlated according to current research. A recent attempt to correlate intestinal flora with tumor immunotherapy revealed that the gut microbiota significantly affect the outcome of tumor immunotherapy, which demonstrates how the body’s systems interact with one another [98].

The effectiveness of the anti-PD-L1 treatment in melanoma was boosted when selected gut microbiota were taken orally, because they raised the degree of the tumor-specific T-cell response, the tumor’s infiltration by CTLs, and IFN production. The effectiveness of immunotherapy was enhanced by the high prevalence of several species among the microbiota. The composition of these species can be seen in Figure 2 [99].

In the near future, it is anticipated that cancer immunotherapy combined with *A. muciniphila,* one of the crucial probiotics in selective microbiota transplantation, will improve clinical outcomes for patients.

## 6. Conclusions

Finding the best probiotic or prebiotic therapy candidates is crucial from a therapeutic perspective. Therefore, understanding the disease pathogenesis and host factors that affect the efficacy of probiotics and prebiotics as well as the individual responses to probiotic and prebiotic therapies is critical. Probiotic or prebiotic modification of the gut microbiota may be a breakthrough strategy for controlling and managing immune-related diseases.

As a possible probiotic that can benefit from gastrointestinal mucin, *A. muciniphila* is intricately related to the host’s metabolism and immunological response. It has the potential to be a therapeutic target in diseases connected to the microbiota, such as cancer and immunological disorders. Human preliminary findings indicate that the oral administration of *A. muciniphila* is safe, but in the near future, further human clinical trials will need to be conducted to clarify this.

On the other hand, *Faecalibacterium*, which was found to produce butyrate, a metabolic product that induces significant anti-inflammatory effects and contributes to intestine epithelial integrity, may be preferentially useful as a probiotic and prophylactic treatment to avoid several immune-related inflammatory diseases.

## Figures and Tables

**Figure 1 microorganisms-10-02382-f001:**
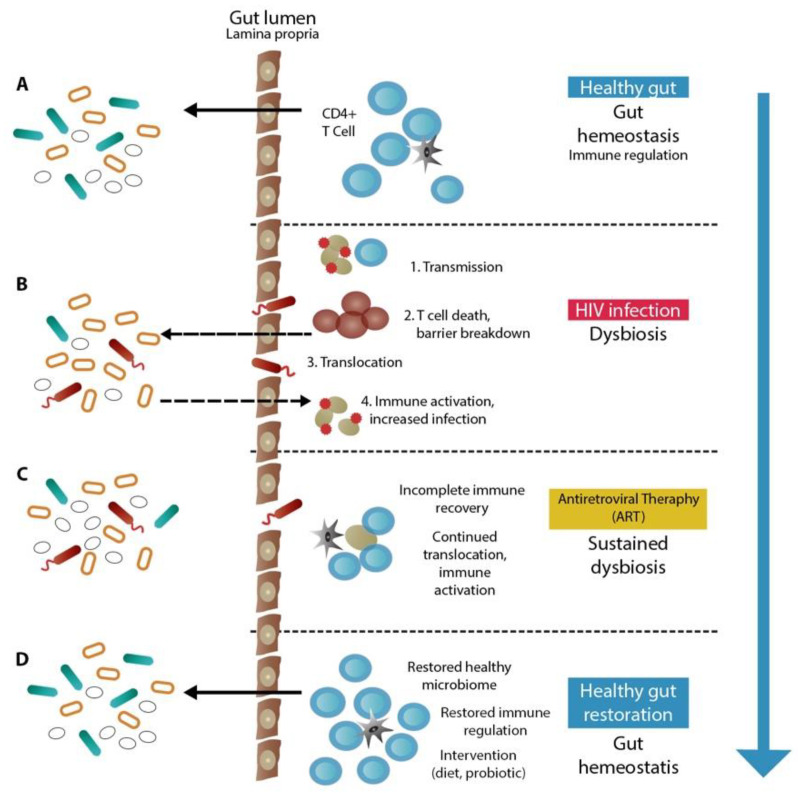
Gut microbiome dysbiosis and immune system in HIV patient. (**A**) Microbiome in normal gut has a role to maintain immune system homeostasis; the dysbiosis caused by a certain condition or disease (**B**) may influence the host immune system. The intervention may provide recovery through relocation of microbiome and immune reactivation (**C**). This results in healthy gut restoration after full immune reconstitution and restored immune regulation (**D**) [73].

**Figure 2 microorganisms-10-02382-f002:**
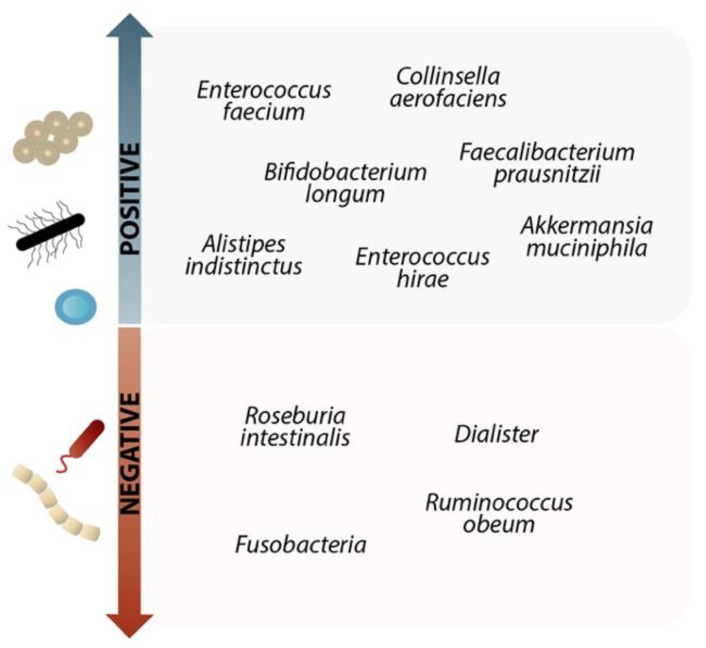
The impact of specific gut microbiomes on anti-PD1 and anti-PD L1 treatment. Due to the high presence of the following species on the “positive” side, immunotherapy is more effective. Administration of these species boosts the quantity of the tumor-specific T-cell response, increasing the amount of IFN-γ produced by the tumor and CTL infiltration, strengthening CTL priming, and enhancing the effectiveness of the anti-PD-L1 therapy. For melanoma patients undergoing combined anti-CTLA-4 and anti-PD-1 therapy, increased *F. prausnitzii* concentration may potentially be advantageous. Before treatment, the anti-PD-1 immunotherapy responses in patients with metastatic melanoma were already overrepresented by *Bifidobacterium longum*, *Collinsella aerofaciens*, and *Enterococcus faecium*. On the other hand, species on the “negative” side reduce the effectiveness of immunotherapy [100,101,102,103,104].

## Data Availability

Not applicable.

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
