# Peer review of "Akkermansia muciniphila and Faecalibacterium prausnitzii in Immune-Related Diseases"

_microorganisms, 2022, doi:10.3390/microorganisms10122382_

Round 1

Reviewer 1 Report

Very timely and well-written overview.

Author Response

Thank you very much for your kind response 

Reviewer 2 Report

Major points

This manuscript generally needs more precise explanation about the mechanism how the genera alter cytokine profiles or NFkB activities and which cell types are targeted.  In several portions, the description is redundant.

English editing by native English writers is required.

Specific pints

Line 146-149 Which is the cell sources for IL-10,4 or TNFa IFNg? 

The effects are mediated by acetate propionate ?

Line 149 The authors showed that A. muciniphila had strong anti-inflammatory and protective effects 

That is rather discrepancy from its effects on cytokines profiles. Please explain the discrepancy.

Line 169 Which is the cell sources for IL-10, 12, IFNg? 

The effects are mediated by butyrate? 

Line 180 The probiotic application route is oral intake? Or microbiota transplantation ?

Line 185 Treg induction is mediated by butyrate?

Line 187 please detail the components of MAM. Does the MAM include butyrate?

Line 215 Which kinds of pathogens’ infections?

Line 230 Low C3 level corresponds to high disease activity. 

Thus Ruminococcus, Bacteroides, and Akkermansia might not improve but exacerbate SLE. 

Please explain the discrepancy.

Line 234 How about the abundance or Faecalibacterium in SLE patients compared to normal controls.

Line 286-8 The correction of Th2 cytokine profiles or restoration of filaggrin by A. muciniphila and F. prausnitzii are mediated by SCFAs?

Line 306 Which cell types are target of NF-κB inhibition? Enterocytes? The inhibition is mediated by butyrate?

Line 330 More specific description is needed for Th type skewing in psoriasis. The enhancement of  th17 and down regulation of Treg in psoriasis.

IL-17A is protective for intestine and it’s role in IBD is different from that in psoriasis. This should also be described. And how  F. prausnitzii might affect individual diseases should be commented.

Line 337 The names of genera used as probiotics should be described 

Line 405 The description ’A. muciniphila in people with ulcerative colitis' fecal microbiome (17)’ is not related to HIV patients and unnecessary.

Line 443 Title ‘Cancer immunotherapy’ is better.

Fig 2 F. prausnitzii induces Treg and might suppress the induction of anti cancer CTL. Thus positive effects of this genus on anti-PD1 and anti-PD L1 treatment is rather discrepant. How do the authors explain the discrepancy?

Author Response

Please see the attachment below for response to reviewer

Round 2

Reviewer 2 Report

I think that Faecalibacterium prausnitzii might induce regulatory T cells which inhibit CTL killing cancer cells. However, in the figure the authors categorizes this species in the positive group inducing anti-tumor immunity. Please discuss if the species enhances or attenuates the anti-cancer immunity in the text.
